

# Snow accumulation and ablation measurements in a mid-latitude mountain coniferous forest (Col de Porte, France, 1325 m alt.): The Snow Under Forest field campaigns dataset

Jean Emmanuel Sicart[1], Victor Ramseyer[1], Ghislain Picard[1], Laurent Arnaud[1], Catherine Coulaud[1], Guilhem Freche[1], Damien Soubeyrand[1], Yves Lejeune[2], Marie Dumont[2], Isabelle Gouttevin[2], Erwan Le Gac[2], Frederic Berger[3], Jean Matthieu Monnet[3], Laurent Borgniet[3], Eric Mermin[3], Nick Rutter[4], Clare Webster[5,6], Richard Essery[7].

[1] Univ. Grenoble Alpes, IRD, CNRS, Grenoble INP, IGE, 38000 Grenoble, France.
[2] CNRM UMR 3589, Météo-France/CNRS, Centre d'Études de la Neige, Grenoble, France
[3] Univ. Grenoble Alpes, INRAE, LESSEM, 2 rue de la Papeterie-BP 76, St-Martin-d'Hères, F-38402, France.
[4] Department of Geography and Environmental Sciences, Northumbria University, Newcastle upon Tyne, NE1 8ST, UK
[5] WSL Swiss Federal Institute for Snow and Avalanche Research SLF, Davos, Switzerland
[6] Department of Geosciences, University of Oslo, Norway
[7] School of GeoSciences, University of Edinburgh, UK

*Correspondence to*: Jean Emmanuel Sicart (jean-emmanuel.sicart@ird.fr)

**Abstract.** Forests strongly modify the accumulation, metamorphism and melting of snow in mid and high-latitude regions. Recently, snow routines in hydrological and land surface models have been improved to incorporate more accurate representations of forest snow processes, but model inter-comparison projects have identified deficiencies, partly due to incomplete knowledge of the processes controlling snow cover in forests. The Snow Under Forest (SnoUF) project was initiated to enhance knowledge of the complex interactions between snow and vegetation. Two field campaigns, during the winters 2016-17 and 2017-18, were conducted in a coniferous forest bordering the snow study at Col de Porte (1325 m a.s.l, French Alps) to document the snow accumulation and ablation processes. This paper presents the field site, instrumentation, and collection methods. The observations include distributed forest characteristics (tree inventory, LIDAR measurements of forest structure, sub-canopy hemispherical photographs), meteorology (automatic weather station and radiometers array), snow cover and depth (snow poles transect and laser scan), and snow interception by the canopy during precipitation events. The weather station installed under dense canopy during the first campaign has been maintained since then and provides continuous measurements throughout the year since 2018. Data are publicly available from the repository of the Observatoire des Sciences de l'Univers de Grenoble (OSUG) data center at http://dx.doi.org/10.17178/SNOUF.2022 (Sicart et al., 2022).





## 1 Introduction

Around 20% of Northern Hemisphere snow overlaps with boreal forest, so sub-canopy snow cover has a key control on eco-hydrological processes (e.g., Rutter et al., 2009). Forests strongly modify the accumulation, metamorphism and melting of snow, they intercept part of the precipitation, modify radiation fluxes and surface roughness, and reduce albedo and wind speed (e.g., Otterman et al., 1988; Pomeroy et al., 2008; Musselman et al., 2012; Essery, 2013). The model inter-comparison project SNOWMIP2 (Essery et al., 2009; Rutter et al., 2009) evaluated 33 forest snow models differing in both process complexity and canopy implementation approaches. Major deficiencies of modeling snow in forests were identified, and the project concluded that model performance was limited by incomplete knowledge of the processes controlling snow cover in forests. Since then, numerous measurement campaigns have been conducted (e.g., Webster et al. 2016, 2018; Malle, et al., 2019; Mazzotti et al., 2019; Hojatimalekshah et al., 2021) and snow routines in hydrological and land surface models have been enhanced to incorporate more accurate representations of forest snow processes (e.g., Ellis et al., 2013; Gouttevin et al., 2015; Boone et al., 2017; Mazzotti et al., 2020). However, these improved routines still represent canopy as one homogeneous layer without accounting for all the effects of particularly vertical canopy heterogeneity on snow accumulation and ablation processes. Detailed snow and meteorological measurements are therefore still required, and remain an important step, to better understand the complex interactions between snow and vegetation. Col de Porte (CDP) is a mid-elevation site located at 1325 m altitude (45.3° N, 5.77° E) in the Chartreuse mountain range in France, with a meadow bordered by a coniferous forest. Morin et al. (2012) and Lejeune et al. (2019) presented the meadow observation site that has been operated since 1959 by CEN-MeteoFrance. Daily measurements of snow depth, air temperature, and precipitation amount have been performed since 1960. Hourly measurements of meteorological and snow variables required to run and evaluate detailed snowpack models such as Crocus (Vionnet et al., 2012) started in 1987 and have been almost continuous during the snow seasons since 1993. CDP is part of several observation networks at the national scale (e.g., Observation pour l'Experimentation et la Recherche en Environnement CryObsClim and Systèmes d'Observation et d'Expérimentation au long terme pour la Recherche en Environnement des glaciers, GlacioClim) and at the international scale (e.g., ILTER European Research Infrastructure, WMO Global Cryosphere Watch CryoNet network, GEWEX INARCH). For more details, the reader is referred to Lejeune et al. (2019). Only a few studies have investigated the snow cover distribution in the forest of CDP (e.g., Durot, 1999); however, the immediate proximity of the forest parcel to the historical, long-term open-area snow observatory of CDP offers a good opportunity to understand and relate the sub-canopy meteorological and snow processes to their open-area counterparts.

Two field campaigns have been conducted in the conifer forest bordering the reference meadow site to document the snow accumulation and ablation processes: from 16 January 2016 to 21 March 2017 and from 1 December 2017 to 15 March 2018. This paper presents the measurement methods that were applied in the forest plot during these two field campaigns. The observations include distributed forest characteristics (tree inventory, LIDAR measurements of forest structure, sub-canopy hemispherical photographs), meteorological variables (automatic weather station and radiometer array), snow height and water equivalent (snow poles transect and laser scan), and transects of snow interception by the canopy during precipitation events.



The dataset also includes continuous measurements from the weather station in the forest from March 2018 to June 2022.
Complementing the datasets, the repository of the Observatoire des Sciences de l'Univers de Grenoble data center also includes
technical information, photographs and a detailed plan of the instrumentation.
**2 Site and forest description**
**2.1 Site**
The study site is a triangular forest parcel of 2000 m$^2$ (Figure 1) next to the meadow where the historical open-area snow
measurements are conducted. It is delimited by a fence along its south and northeast sides. Its west side corresponds to the
edge between the forest and the open meadow area. The terrain slope is around 10° oriented toward east-north-east. The stand
is dominated by Norway spruce (*Picea abies*). Young silver firs (*Abies alba*) are present, mainly in the western part of the
parcel. Some broadleaved trees are located along the west edge. The parcel exhibits two gaps in the canopy. The smallest one
is in the south-west, while the larger one is at the center and extends toward the south fence (Figure 1). During the first [second]
campaign, the annual maximum snow depth was around 100 cm [160 cm] in the open site (meadow reference site) and only
around 50 cm [130 cm] under the canopy.



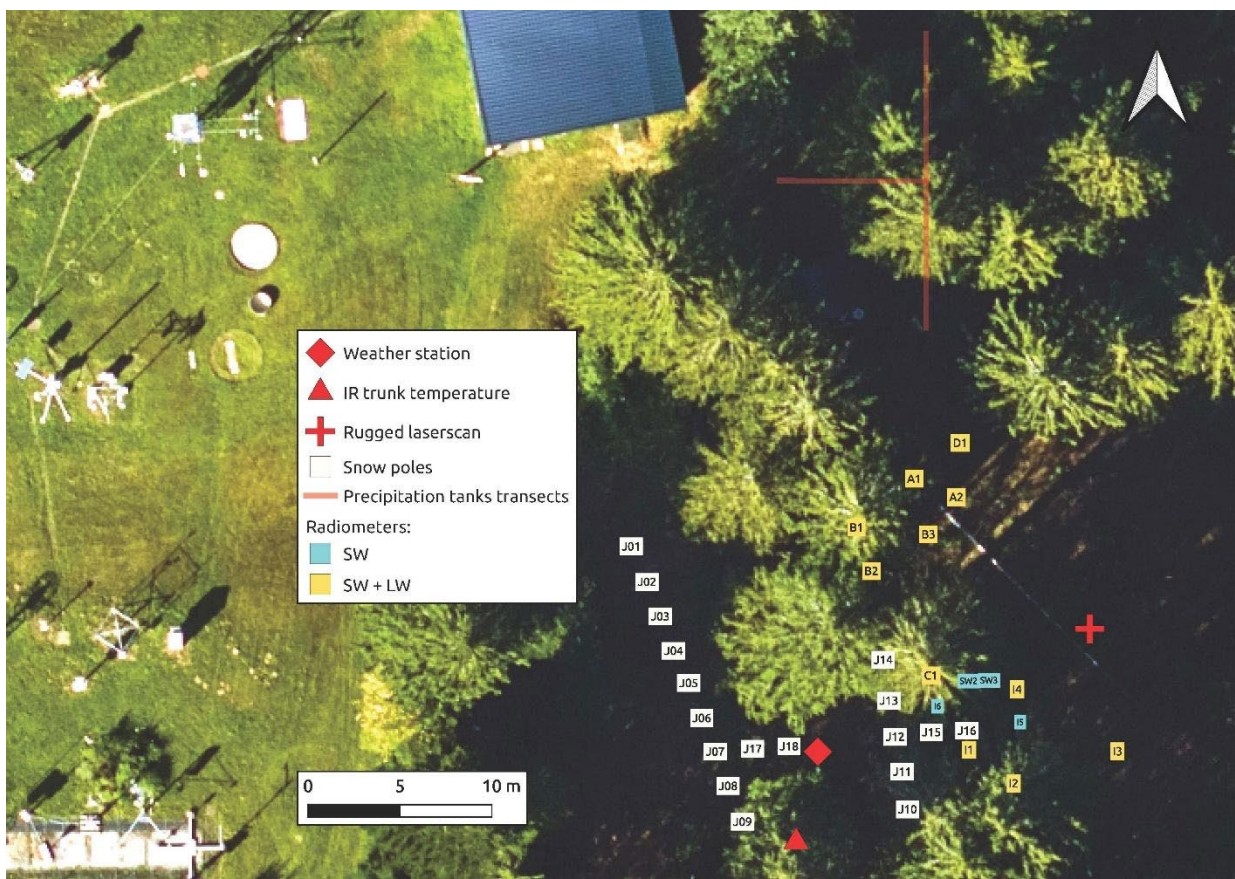

Figure 1: Aerial photograph of the site with locations of the sensors. The sensors of the open meadow area appear on the left of the picture.

## 2.2 Forest measurements

### 2.2.1 Forest inventory

An inventory of trees higher than 1.3 m took place during field campaigns between September 2016 and July 2018. On 13-14 September 2016, live and dead trees were inventoried, and the following observations and measurements were performed:

- diameter at breast height (DBH, measured with a tape measure at 1.3 m height above the ground, upslope of the tree),
- species,
- tree height measured with a Haglöf Vertex 4 hypsometer, only for trees with a DBH larger than 7.5 cm.

Tree identification numbers were painted on the trunk at a height around 1.3 m above ground. Three reference poles were positioned in the site and geolocated using a Trimble GeoExplorer 6000 XH GNSS receiver. Trees were mapped relative to a nearby pole by measuring the ground distance (Vertex 4 hypsometer), slope (clinometer) and azimuth (compass). Tree position



precision relative to the reference pole is expected to be better than 50 cm, whereas GNSS precision under the forest canopy
is of the order of a few meters. 141 trees were inventoried: 128 live trees, 3 dead trees and 10 stumps.
On 5 May 2017, vertical crown projections of live trees with a DBH larger than 7.5 cm were measured with a tape measure as
the horizontal distances between the trunk center and the vertical projection of the furthest live branch along north, south, east
and west directions. If several tree stems were sprouting from a common base, the whole clump was considered to have one
single crown and its extension was measured from the stem with the largest diameter in the clump.
On 20 June 2017, tree positions were measured with a Leica TS02 total station located in the open area at the west of the forest.
The total station position was recorded with a Trimble R2 differential GNSS receiver, ensuring centimetric accuracy.
On 27 July 2018, heights and crown extensions were measured on trees with a DBH smaller than 7.5 cm. The tree inventory
was extended outside the southern fence to include trees which might cast shadows inside the forest parcel (DBH, height,
crown extension, species). Their positions were measured with slope (clinometer), azimuth (compass) and ground distance
(Vertex 4 hypsometer) relative to a reference pole located with a GNSS receiver.
Tree easting and northing values in the RGF 93 - Lambert 93 projected coordinate system were then derived from the total
station coordinates if available, or from their polar coordinates relative to a reference pole. Tree altitude values were computed
from the airborne laser scanning data (see Section 2.2.2) by bilinear interpolation of the ground-classified points at the location
of trees. Figure 2 shows a map of inventoried live trees and canopy heights. In the forest stand inside the fence, most of the
trees are between 30 and 40 m high (Figure 3) and the total basal area is about 66.3 m2/ha. It includes 52 firs, 43 spruce and
33 broadleaved trees. Trees measured outside the southern fence are 9 spruce, 2 firs and 7 broadleaves.
Tree position accuracy is estimated to be better than 10 cm for the trees measured with the total station inside the fenced area
(it is estimated to be around 50 cm for additional trees outside the fence). Luoma et al. (2017) reported a precision of 0.5 m
(standard deviation) for height measurements with a Vertex 4 clinometer. Elzinga et al. (2005) reported a standard deviation
of 0.5 cm for diameters measured with a tape measure. Measurement errors on crown extension is mostly due to the difficulty
to assess the vertical projection of the branches' extent on the ground. Accuracy is expected to be from 10 cm for small trees
(height smaller than 4 m) to 50 cm for the tallest ones (around 30 m).

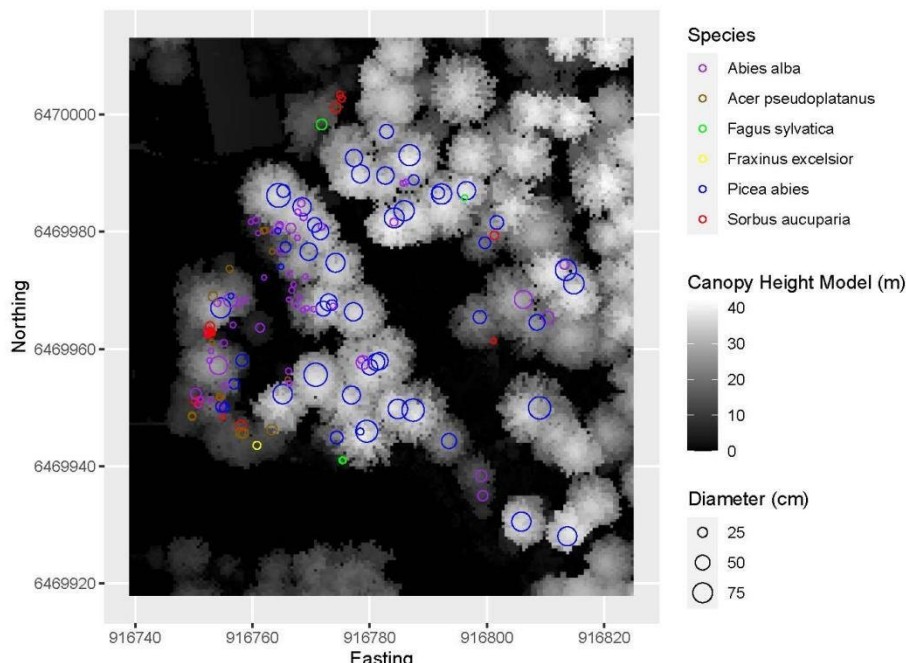

116              Figure 2. Map of inventoried live trees and canopy height model derived from airborne laser scanning.

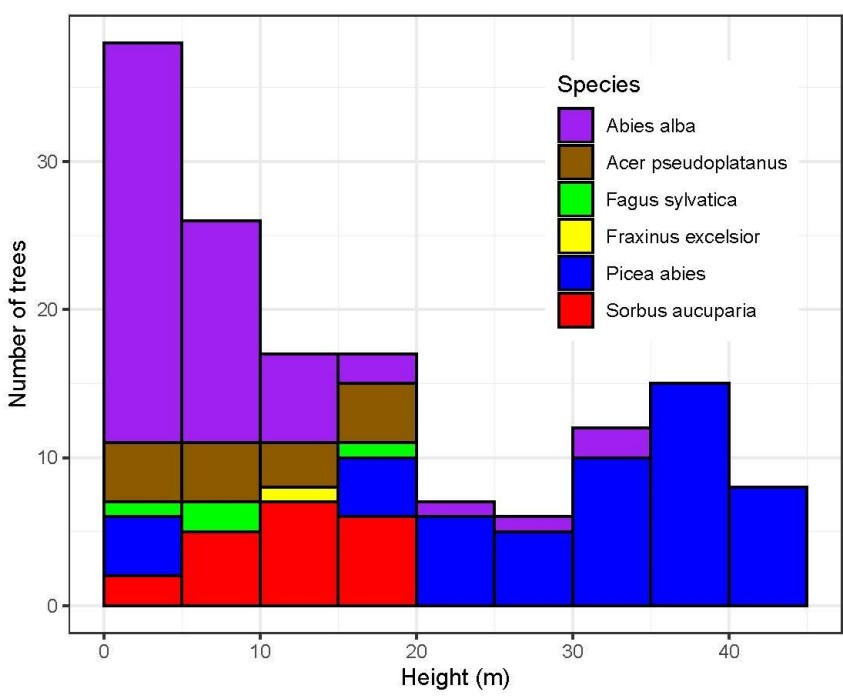

118                              Figure 3: Height distribution of living trees.




*2.2.2 Airborne remote sensing measurements*
Airborne laser scanning (ALS) is a remote sensing technique based on LIDAR which can provide a 3D point cloud of forest
structure. The geometric information on the vegetation can be processed to derive forest metrics used to parametrize snow
interception (e.g., Helbig et al., 2020). ALS was acquired during a campaign covering 123.5 km$^2$ between 30 August and 2
September 2016, using a Riegl LMS Q680i sensor mounted on a helicopter. The scheduled flight height and speed were 750 m
above ground and 70 knots, respectively. The scan frequency was 300 kHz with a scan angle of ± 30°. The aircraft trajectory
was computed from the Inertial Measurement Unit and GPS data. Point coordinates were extracted and computed using the
RiAnalyse and RiWorld software. The point cloud was then classified as ground / non-ground using Terrasolid.
To assess the accuracy on elevation measurements, 318 ground control points were measured with differential GNSS in 11
flat, vegetation-less plots. Differential GNSS accuracy is around 2-3 cm in such areas. Comparison of the control points with
the point cloud yielded to an altitudinal accuracy of 4.7 cm (root mean square error of differences), with a bias of -0.3 cm.
The point cloud was delivered in tiled LAS 1.2 files. The coordinate system was RGF 93 - Lambert 93, with altitude in the
system NGF-IGN69. The point cloud corresponding to the study site with a 200 m buffer was exported in a single compressed
LAS file (v1.1 format 1). Pulse density in the study site is 17 points/m$^2$, resulting in densities of 3.3 points/m$^2$ for ground points
and 28 points/m$^2$ with multiple returns for canopy points. The area located more than 30 m to the southeast of the study area
was not covered by the acquisition.
For the extent of the inventoried trees plus a 30 m buffer, digital surface models were computed at 0.5 m resolution from the
ALS point cloud. The digital terrain model (DTM) was computed by estimating the altitude of each cell center by bilinear
interpolation of ALS points classified as ground. The point cloud was normalized by subtracting the ground altitude at the
position of each point, estimated by bilinear interpolation of ground points. A canopy height model (CHM) was computed by
retaining the highest value of normalized heights in each cell. Cells without values were filled by the median of their 3x3
neighborhood. The DTM and CHM were delivered as raster files in tif format. Aerial photographs were also taken during the
ALS acquisition. Pictures were used to produce a 10 cm resolution RGB orthophoto provided as a tif file for the extent of the
DTM. Figure 4 shows a perspective view of the 3D point cloud acquired by the airborne laser scanning.

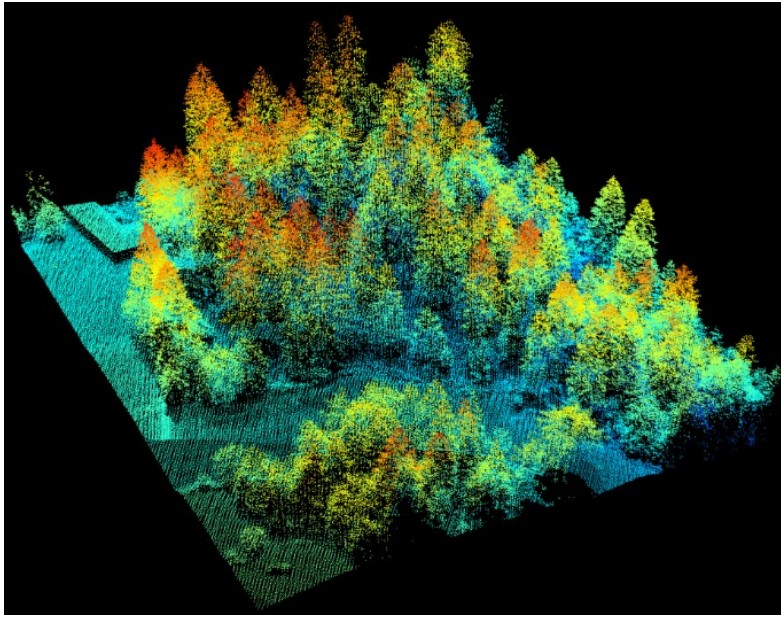

Figure 4: Perspective view of the 3D point cloud acquired by airborne laser scanning. Points are colored according to their altitude.

### 2.2.3 Hemispherical photographs

Hemispherical photographs were taken on 4 September 2017 at the weather station and at each radiometer in homogeneous overcast conditions to ensure spatially uniform sky brightness. A Nikon Coolpix 4300 digital camera was used with a Nikon FC-E8 fisheye lens, mounted 60 cm above the ground surface on a tripod. The camera was aligned to north with a compass and carefully levelled using a bubble level. There was no snow in the canopy. Sky view factors were then calculated following Essery et al. (2008) assuming an equiangular lens projection. To distinguish vegetation from sky pixels, and to calculate the sky view factor at each location, a brightness threshold was visually adapted to each hemispherical photograph (Figure 5). This allows us to account for variations in illumination conditions during changes in cloud cover or thickness. As a result, the calculated sky view factor ranges from 0.15 to 0.35 at the radiometer sites, mostly situated under a rather dense canopy. Reid et al. (2014) estimated the uncertainty in the sky view factor using this method to be ±0.02.

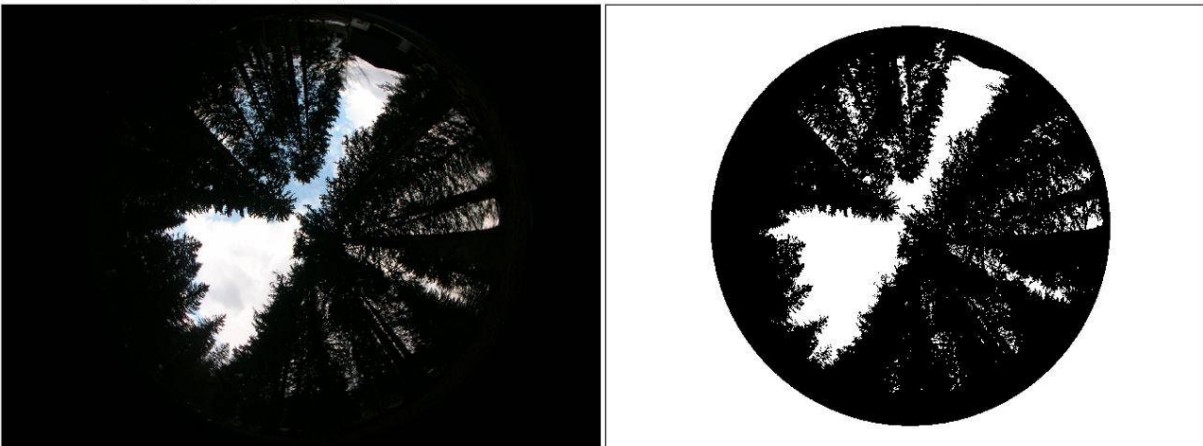

Figure 5: Example of a hemispherical photograph (left) and a binary image (right, calculated sky view factor 0.25). The photograph was taken on 4 September 2017 at radiometer A1 (Figure 1).

## 2.3 Meteorological and snow observations in the forest

### 2.3.1 Radiometer array

An array of 13 CMP3 Kipp & Zonen pyranometers and 11 CGR3 Kipp & Zonnen pyrgeometers was deployed on the snow surface from dense canopy to an opening (Figure 1). During the first campaign, each radiometer was positioned horizontally with a bubble level on a wooden board placed on the snow surface. During the second campaign, the horizontal support of the radiometers was attached to a vertical bar fixed in the ground. The height of the support was adjusted to the snow surface every two or three days. This system allowed better stability and levelling of the radiometers. One-minute averages of the incoming radiation fluxes measured at five second time intervals were recorded by two Campbell Scientific CR3000 data loggers. Inter-calibration of the sensors before the campaigns led to estimates of sensor accuracies close to those announced by the manufacturer: $\pm 12$ W m$^{-2}$ for solar radiation and $\pm 8$ W m$^{-2}$ for longwave radiation, in accordance with uncertainty estimations from similar sensors of Halldin et Lindroth (1992), Philipona et al. (2001), Michel et al. (2008), or Van den Broeke et al. (2004). The radiation data were carefully post-processed to remove periods when the sensors were snow covered or tilted.

### 2.3.2 Weather station

The weather station was installed under rather dense canopy during the first campaign and has been maintained since then. Table 1 lists the sensors installed on the station, their specifications and their accuracy according to the manufacturer. The ultrasonic depth gauge measures the snow height. Ten temperature probes buried in the ground are used to estimate the heat conduction flux. 15-minute averages of the data measured at 10 second time intervals are recorded by a Campbell Scientific CR3000 data logger. An AXIS M1125-E camera took pictures of the surface around the weather station every three hours



during daytime. These images are used to monitor surface and sensor conditions. A Campbell Scientific IR120 infrared sensor
was used to measure the surface temperature of a trunk close to the meteorological station (Figure 1). One-minute averages of
longwave irradiance measured at five second time intervals were recorded by a Campbell Scientific CR1000 data logger.

Table 1: variables measured by the weather station below the canopy along with the sensor type, heights and precision
according to the manufacturer.

| Quantity | Sensor Type | Height (cm)[1] | Accuracy according to the manufacturer |
|---|---|---|---|
| Air temperature, °C and relative humidity, % | Campbell CS215C | 210 | ±0.2 °C<br>±2% in [0-90%]<br>±3% in [90-100%] |
| Wind speed and direction, m s$^{-1}$ and deg. | Gill windsonic | 210 | ±0.3 m s$^{-1}$<br>±3 deg |
| Incident and reflected short-wave radiation, W m$^{-2}$ | Kipp & Zonen CM3 $0.3<\lambda<2.8$ µm | 100 | ±10% for daily sums |
| Incoming and outgoing long-wave radiation, W m$^{-2}$ | Kipp & Zonen CG3 $5<\lambda<50$ µm | 100 | ±10% for daily sums |
| Surface elevation changes, mm | Ultrasonic depth gauge Campbell SR50 | 180 | ±1 cm |
| Temperature in the ground, °C | 108 Campbell | -2.5, -8, -15, -30, -60[2] | ≤ ±0.01°C |

[1] height above snow-free ground
[2] two sensors for each depth

As an example of the use of these measurements, Figure 6 shows hourly changes in air and soil temperatures and snow depth
from December 2018 through May 2019, a winter characterized by deep and sustained snow cover. Snow cover began to build
in late December, reaching a maximum of 90 cm in early February. Melt rates became significant as air temperatures remained
consistently above 0°C through late February. Snow cover disappeared by the end of March, although there were a few
snowfalls through May. Snow cover strongly affected soil temperatures to a depth of 60 cm. The disappearance of the main
snow cover at the end of March suddenly reversed the temperature gradient in the soil. Soil temperatures were also affected
by a few snowfalls in April, associated with short periods of cold weather.

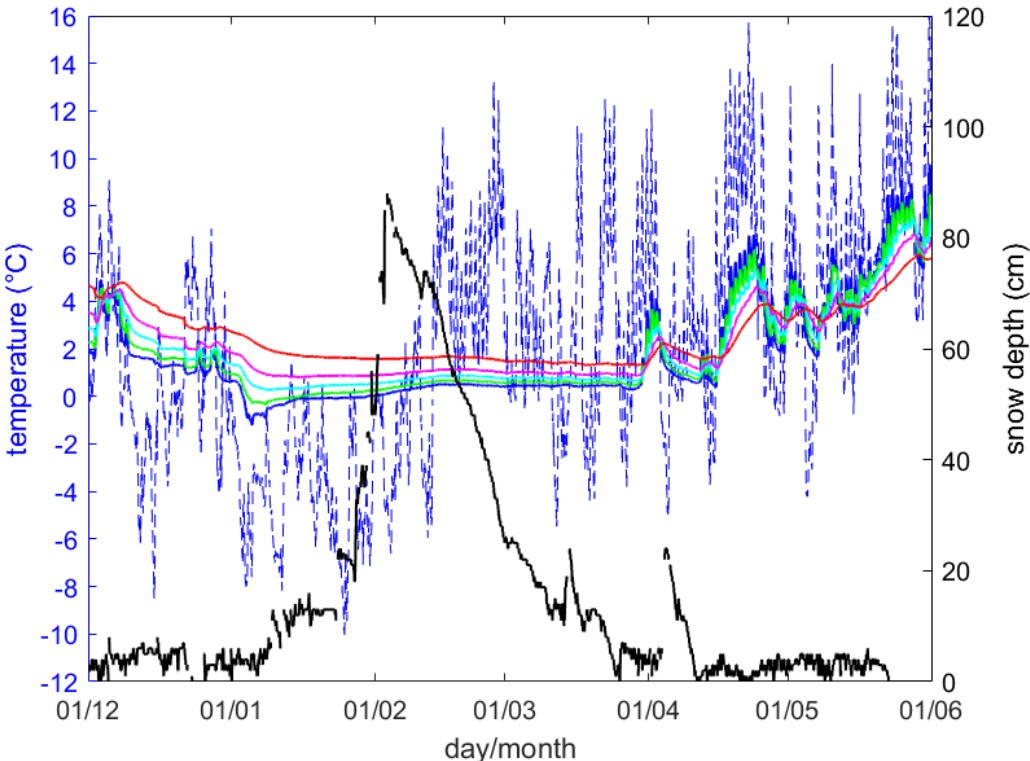

Figure 6: Left Y axis: air temperature (blue dotted line) and soil temperature at 2.5, 8, 15, 30 and 60 cm depth (blue, green, cyan, magenta and red solid lines, respectively). Right Y axis: snow depth (black line). Hourly data from the weather station in the forest (Table 1) from 1 September 2018 to 1 June 2019.

### 2.3.3 Snow measurements

In order to document the spatial variability of snow cover in the forest, a transect of 18 snow poles was deployed in early winter 2016-2017 (Figure 1). The locations of these snow poles (spaced 2 m apart) were georeferenced in Lambert 93 coordinates. The snow height was measured approximately every two weeks during the two field campaigns. Snow water equivalent measurements were carried out every week for only four poles at a time, alternating among the 18 poles, to minimize destruction of the local snowpack structure. Detailed studies of Morin et al. (2012) and Lejeune et al. (2019) estimated the uncertainties on snow depth and snow water equivalent measurements to be ± 1 cm and ± 5 %, respectively, in agreement with the estimation of López-Moreno et al. (2020) derived from a comparison of measurements with different snow core samplers. Simultaneously, measurements of snow height and water equivalent were made in the reference meadow site as described in Lejeune et al. (2019).

### 2.3.4 Precipitation tanks



The amount of snow held in a forest canopy can be large and remains difficult to measure. Due to the sublimation of intercepted
snow, a large portion of the snow retained in the canopy never reaches the ground, and the interplay of interception, sublimation
and delayed deposition on the ground creates significant below-forest heterogeneity in snow accumulation (e.g., Helbig et al.,
2020). To try to measure snow interception by the canopy, 24 "precipitation tanks" (1 m x 0.39 m) were built and then deployed
under the canopy in three eight-meter transects at the start of winter 2017-2018 (Figures 1 and 7). The uncertainties of this
new measurement method developed by the CEN is difficult to estimate. Vincent (2018) estimated the measurement
uncertainty at about 5%, but additional studies are required to specify it. The mass of snow collected by the precipitation tanks
was measured seven times from 20 February to 3 April 2018.

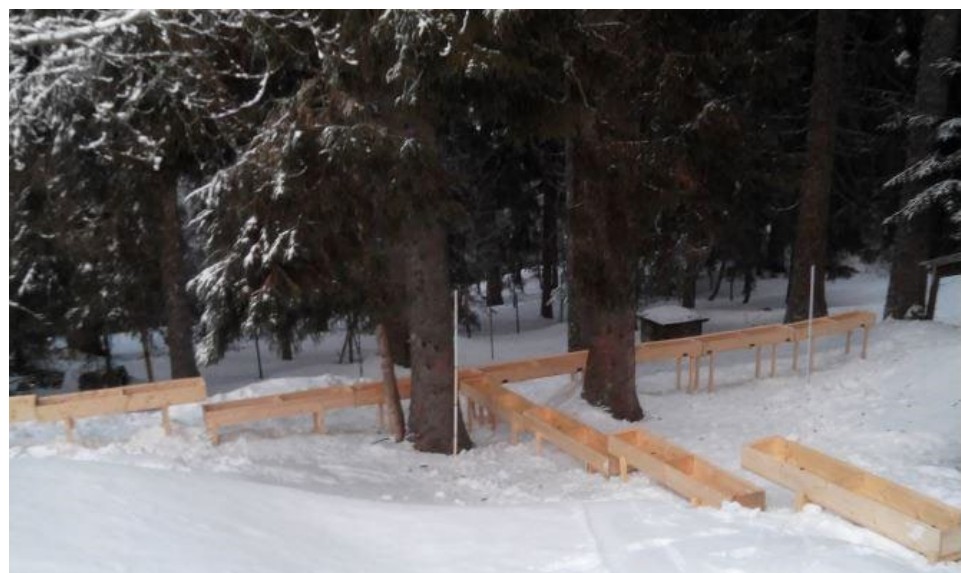

227                      Figure 7: Precipitation tanks installed below the canopy (photograph by Y. Lejeune).


*2.3.5 Rugged Laser Scan*
The Rugged Laser Scan (RLS) is a scanning laser meter that was installed at about 4 m above the ground close to the center
of the main clearing to monitor spatio-temporal variations of snow depth under various canopy covers on a daily or two-daily
basis (Figure 1). The device is described in detail in Picard et al. (2016) and is specially designed to monitor snow heights. It
comprises a laser meter mounted on a 2-axis stage and can scan ≈ 200000 points in 4 hours. The laser meter was used in scan
mode. With a setup at 4 m height, and azimuth angles varying from -90° to +90° and zenith angles varying from 19° and 62°,
the scanned area is a half-disk of radius ~7 m, with a surface area of about 80 m². The area encompasses three pairs of
radiometers installed on the snow surface. Data acquired by the laser meter for a given day are processed to build a cloud of
x, y, z points, which is then interpolated and averaged on a regular 3-cm grid. The grid is common to all measurement days so
it is easy to compare the evolution of the snow surface. The vertical precision was evaluated to be about 3 mm and the accuracy
to be 1 cm (Picard et al. 2016).
The RLS was operated during the two field experiments. The first season was from 22 February 2017 to 4 April 2017 (42
days) and had 42 valid acquisitions (once a day). The second season was from 5 December 2017 to 11 March 2018 (160 days)
and has 81 valid acquisitions because scans were scheduled every other day during the winter (accumulation period) and every
day during the melt season. Figure 8 shows an example of snow depths on 15 April 2018 and Figure 9 shows the changes in
daily spatial averages of the snow depth during the 2017-18 field campaign.

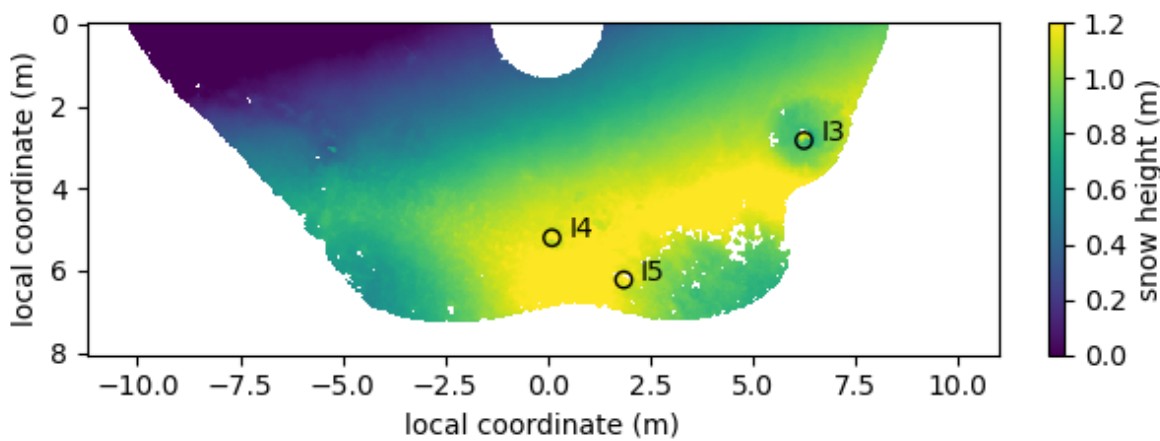


Figure 8: Snow depths (m) derived from the digital terrain models (DTM) measured by the rugged Laserscan, obtained by
subtracting the snow-free DTM from the 15 April 2018 DTM. The measurement area encompasses three radiometers (I3, I4
and I5).


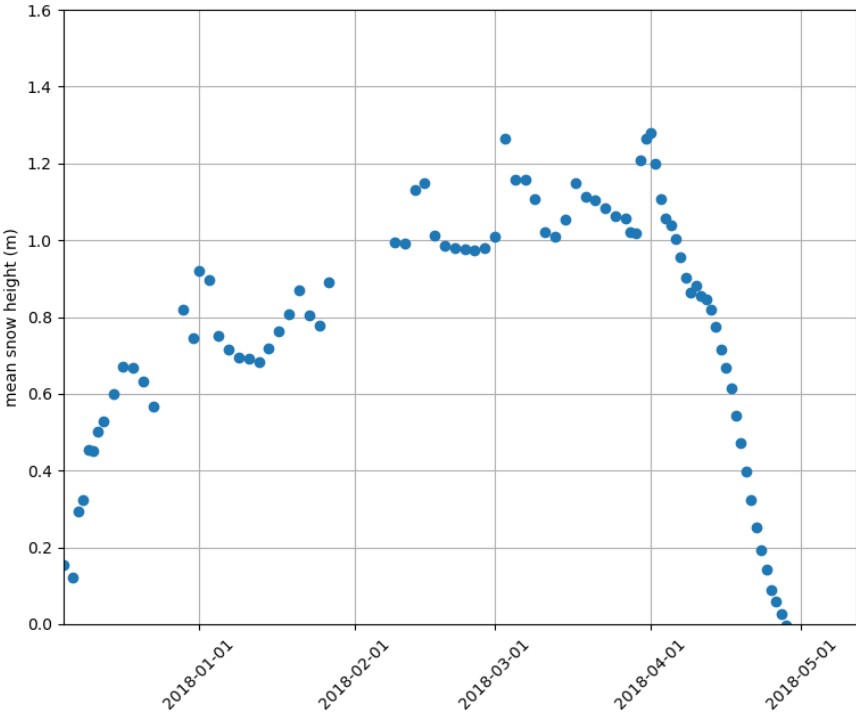

Figure 9: Daily spatial averages of the snow depth measured by the rugged Laserscan during the 2017-18 field campaign.

## 3 Spatial variability, measurement uncertainties and data validation

This section provides estimations of the dataset uncertainties related to measurement uncertainties and spatial variability of the variables within the measurement plot. The uncertainties on the sensors and on the measurement methods have been described in the previous sections. For meteorological measurements, sensor manufacturers generally provide reliable information on sensor accuracy (Table 1). In this Section, comparisons of radiation, air temperature, and snow measurements at different locations provide a better insight into the measurement uncertainties and a first validation of the data set.

Figure 10 illustrates the spatial variability of the incoming shortwave and longwave radiation fluxes below various forest covers. It shows the effects of clouds and canopy cover on the sub-canopy 15-min radiation fluxes during an overcast day and a clear sky day of the 2017 campaign. Under thick cloud cover (January 31), shortwave radiation, mostly diffuse, reaching the ground remains small but steadily increases with decreasing canopy cover (increasing sky view factor). Sky and vegetation were characterized by similar temperature and longwave emittance (both close to 1), and all the pyrgeometers recorded similar longwave radiations fluxes (within a few of W m$^{-2}$, confirming the good accuracy of the sensors), without relation with the canopy cover. In clear sky (February 18), shortwave irradiance is mostly direct. Sun flecks on the ground below the canopy

caused peaks of shortwave irradiance of various amplitudes and at different times at the different pyranometers, superimposed
on the spatially constant diffuse shortwave radiation that has penetrated through the canopy. The diurnal changes of sub-
canopy longwave irradiance are remarkably parallel between the different measurement sites. The constant offset between the
signals is related to the canopy cover due to the large contrast between the large emittance of vegetation and the small emittance
of clear, cold sky: the larger the sky view factor, the smaller the longwave irradiance.

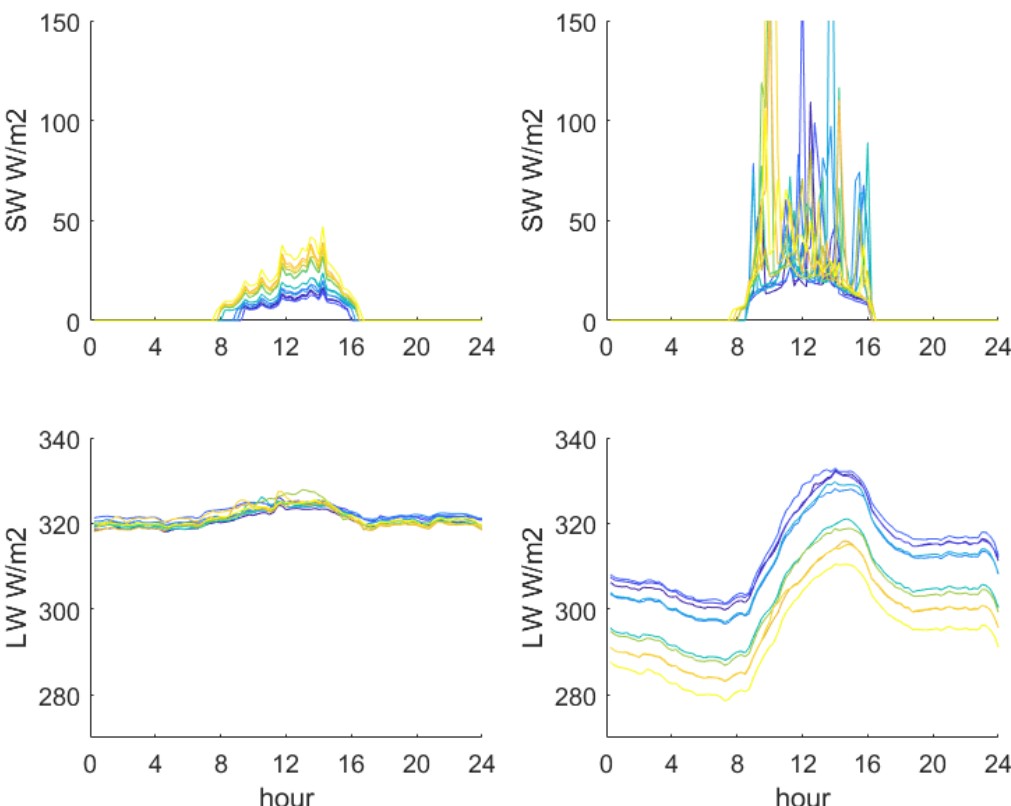


Figure 10: Shortwave (a, b) and longwave (c, d) incoming radiation fluxes measured by each radiometer during an overcast
day (January 31: a, c) and during a clear-sky day (February 18: b, d). 15-min averages of the sub-canopy fluxes during the
2017 campaign. Line color is related to the sky view factor $V_f$ from dark blue (lowest $V_f = 0.17$) to yellow (highest
$V_f = 0.32$).

Figure 11 illustrates the hourly air temperature differences between forest and meadow. During daytime, the forest generally
is a few degrees colder than the open meadow site, with the difference increasing on clear sky days when air is the warmest
(high $T_{open}$). During the night, the forest is generally slightly warmer than the meadow, with the difference reaching a few
degrees on cold clear-sky nights (low $T_{open}$). Thus, on average, air temperature is quite similar in the forest and the meadow
site ($dT \sim 0.2$°CK on average during the 2018 campaign): warmer nights counterbalance cooler days in the forest relative to
the meadow. In addition, warmer cloudy periods tend to counterbalance cooler clear-sky periods in the forest relative to the
meadow.

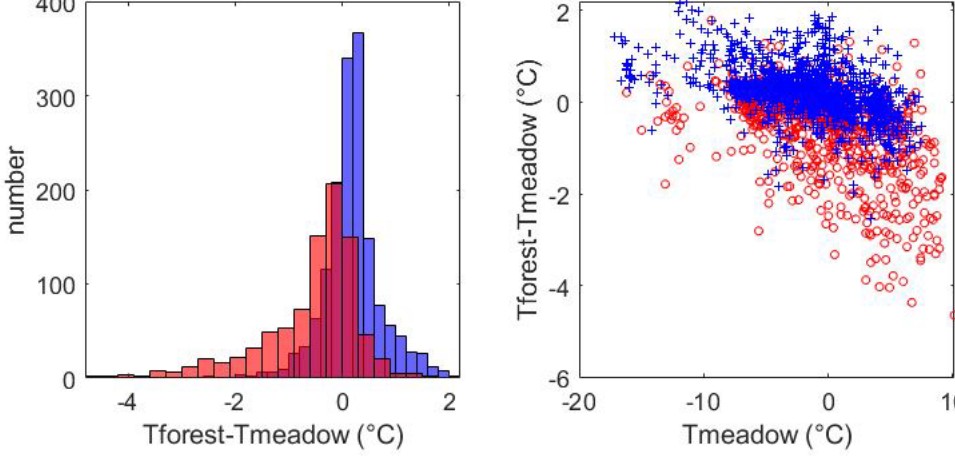

Figure 11: difference of hourly air temperature between forest and meadow ($T_{forest}$-$T_{meadow}$) during the 2018 campaign. The
distinction between daytime (red bars and circles) and nigthtime (blue bars and crosses) values is based on a threshold on the
shortwave incoming radiation fluxes in the meadow site ($SW < 10$ W m$^{-2}$ during nightime).
Figures 12 and 13 illustrate the spatial variability of snow depth and snow water equivalent measurements in the open meadow
and along the snow pole transect in the forest (see locations in Figure 1) during the 2017 and 2018 field campaigns. As
previously mentioned, snow cover lasted several weeks longer and was deeper in the second campaign than in the first, reaching
a maximum in the meadow of 160 cm and 100 cm, respectively. The seasonal maximum snow depth under the canopy was
smaller than that of the meadow by factors ranging from 0.20 to 0.75, depending on the local canopy cover. For the snow water
equivalent, these ratios ranged from 0.16 to 0.60. Relative decreases in snow depth and water equivalent in the forest transect
compared to the meadow were greater during the first campaign characterized by shallow snow cover. Figures 12 and 13
suggest that the effects of the forest on snow cover are more marked during the winter accumulation season (likely due to
interception of snow by the canopy), whereas melt rates during the short ablation season appear to be quite similar in the forest
and the meadow. However, further analysis of snow and meteorological data is required to investigate this point.

Earth System
Science
Data

Open Access | Discussions

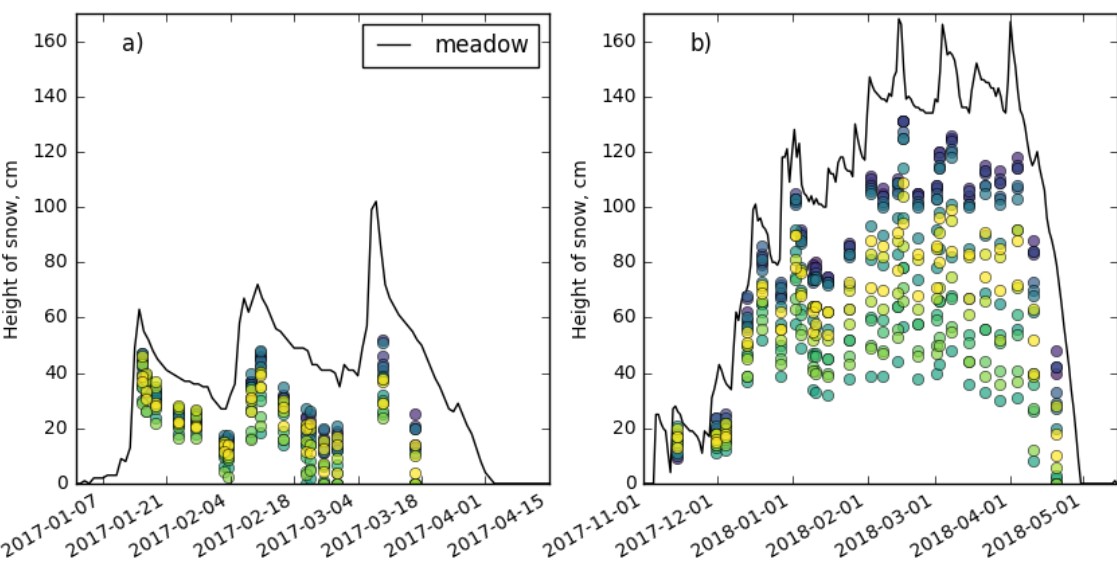

Figure 12: Reference snow depth measured in the meadow from Lejeune et al., 2019 (black line) and manually measured at the snow poles transect in the forest (circles, each color corresponds to a snow pole) during the 2017 (a) and the 2018 (b) field campaigns.

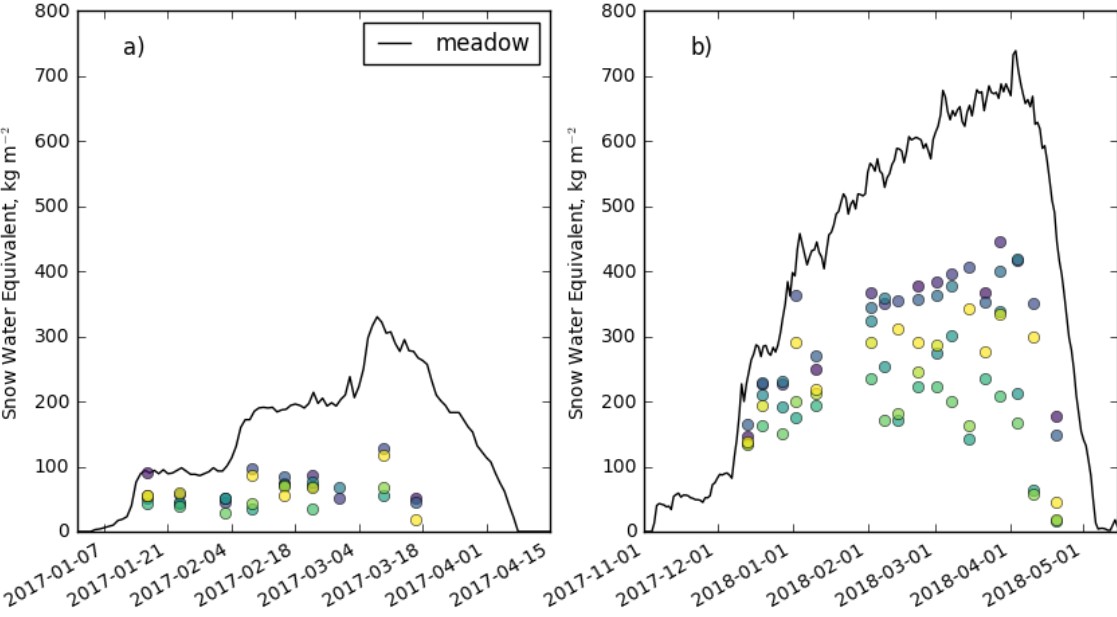

Figure 13: Reference Snow Water Equivalent measured in the meadow site with cosmic ray neutron sensor (black line, see details in Lejeune et al., 2019) and manually measured at the snow poles transect in the forest (circles, each color corresponds to a snow pole) during the 2017 (a) and the 2018 (b) field campaigns.




Table 2: Link to the dataset repository.

| Data set | Period | Format | Repository |
|---|---|---|---|
| Forest inventory | 13-14 September 2016<br>27 July 2018 | csv | https://doi.osug.fr/data/public/SNOUF/forest/ |
| Hemispherical photographs | 4 September 2017 | png | https://doi.osug.fr/data/public/SNOUF/hemis-photos/ |
| Rugged laser scan | 22 Feb to 4 April 2017<br>5 Dec 2017 to 13 May 2018 | netCDF | https://doi.osug.fr/data/public/SNOUF/laser-scan/ |
| Airborne laser scanning | 30 August and 2 September 2016 | 100 | https://doi.osug.fr/data/public/SNOUF/lidar/ |
| Weather station and radiometer array measurements | 16 Jan 2016 to 14 June 2022 | csv | https://doi.osug.fr/data/public/SNOUF/meteo/ |
| Snow pole and precipitation tank measurements | 16 Jan 2016 to 21 March 2017<br>1 Dec 2017 to 15 March 2018 | xls | https://doi.osug.fr/data/public/SNOUF/snow/ |


**3 Conclusions**
The datasets collected in the Col de Porte coniferous forest will allow research on the effects of the canopy on snow
accumulation and ablation processes under different canopy covers. Two intensive field campaigns were conducted during the
winters of 2016-17 and 2017-18 and an automatic weather station has been maintained under the canopy since then.
Meteorological and snow measurements (automatic weather station, radiometer array, snow pole transect, laser scan,
precipitation tanks to estimate snow interception by the canopy) were complemented by canopy observations (tree inventory,
LIDAR measurements of forest structure, sub-canopy hemispherical photographs). Continuous measurements throughout the
year at high temporal frequency (15-minute) from the meteorological station allow hydrological and ecological studies related
to seasonal changes in micrometeorological and soil conditions.
**Data availability**
All datasets described and presented in this paper can be openly accessed from the repository of the Observatoire des Sciences
de l'Univers de Grenoble (OSUG) data center at: http://dx.doi.org/10.17178/SNOUF.2022 (Sicart et al., 2022). Table 2
provides the links to the different datasets.



## Author contributions

JES organized the data and wrote the first draft of the manuscript. JMM and YL cleaned and corrected the forest and snow measurements, respectively. LA and GP cleaned and corrected the laser scan measurements. VR and DS analyzed the meteorological data. All authors participated to the field campaigns, collected and assembled data records, and contributed to writing the paper.

## Competing interests

The authors declare that they have no conflict of interest.

## Disclaimer

Any reference to specific equipment types or manufacturers is for informational purposes and does not represent a product endorsement.

## Acknowledgments and funding

This project was conducted within the grant Labex OSUG@2020 ANR10 LABX56 UGA and with financial support from the IGE and the CEN, through a collaboration between the French institutes IGE, CEN and INRAE, and the universities of Edinburgh and Northumbria in the UK.

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
