# Peer review of "Snow accumulation and ablation measurements in a mid-latitude"

_Earth System Science Data, 2023_

## Author Comment (AC1)

General comments

Dear authors, congratulations for this interesting and well-written article! It will serve the scientific community with a valuable dataset of high quality and a detailed description of the data. The presented dataset is both unique and useful and well comprises the existing data sets for the Col de Porte open site. Accuracies of the measurements are well provided. The text is well structured and of appropriate detail and length. The English is very good and there are only few minor language errors in the text.

The published dataset can be rated significant - in terms of uniqueness, usefulness and completeness. Both data and presentation qualtity are very good.

I only have the following minor specific comments and suggestions for technical improvement. All in all my recommendation is "minor revision", but all this can be done with modest additional effort.

*Thank you for your review. We have taken into consideration all your comments, and a point-by-point response follows.*

Specific comments and technical corrections
- 70: since the tree species are listed later (in Figure 3) it could be sufficient here to formulate it in a more general way and distinguish only between coniferous and deciduous trees
*We feel that the present formulation provides a concise description of both the size and spatial distribution of the three main species of the forest (spruce, fir, broadleaves). Then, it makes it easier for the reader to consider the spatial information of figure 2 and the height distribution information of figure 3.*

- 71: "The smallest one" should probably better be "The smaller one" (if there are only two?)
*Changed*

- Figure 1: (i) caption: the sentence "The sensors of the open meadow area appear on the left of the picture" would better be something like "On the left is the meadow with part of the long-term instrumentation for the Col de Porte site"?
*Changed*
(ii) legend: better move to one of the left corners of the picture so that it does not cover trees. Same for the scale bar and the N arrow.
*Following the reviewer's suggestions, figure 1 has been modified and now has the same extent as figure 2.*
(iii) Rugged laserscan: is it possible to indicate the viewing direction and the scanned area of Figure 8 here (see also comment to Figure 8)?
*As the figure shows a lot of information, we think it's difficult to add new information here. Further information on the area scanned by the RLS has been added in paragraph 2.3.5 and figure 8.*

- Figure 2: (i) can you adopt the extent of this map to exactly the same as the picture in Figure 1, or alternatively, provide a map inlet that shows the two map extents? (I consider the first solution the better one). This would support to identify the location of the devices which are shown in Figure 1 also in Figure 2. (ii) can you indicate the gaps here? Which one is where? (iii) add scale bar and N arrow.

*Figures 1 and 2 now have the same extent. The gaps are now indicated in Figure 2, and a scale bar and N arrow have been added.*

- Figures 1-4: provide corresponding date in the caption
*Done*

- 122: "geometric information on": can you formulate explicitly what can be processed and replace "information on"?
*Changed to "The point cloud can be processed to derive forest metrics used to parametrize snow interception, such as leaf area index or canopy closure"*

- 123: "a campaign covering 123.5 km2": where is this area, does it fully integrate the Col de Porte site?
*Yes, this area fully integrates the Col de Porte site. This information has been added in the text.*

- 127: software packages RiAnalyse, RiWorld and Terrasolid: can you provide version numbers and links?
*Unfortunately, version numbers were not provided by the company in charge of this part of the processing. Links to the software web pages have been added in the text.*

- 128: "accuracy on elevation measurements" should probably be "accuracy of elevation measurements"
*Changed*

- 129: "GNSS accuracy is around 2-3 cm": horizontal or vertical?
*"horizontal" added in the text*

- 130: "yielded to an altitudinal accuracy": delete "to"
*done*

- 131: what is a LAS 1.2 file?
*The text now specifies: "The point cloud was delivered as tiled files in LAS format, which is the most common format for LIDAR point clouds exchange and is maintained by the ASPRS (https://www.asprs.org/divisions-committees/lidar-division/laser-las-file-format-exchange-activities)"*

- 132: with a "200 m buffer": what does this mean? Maybe that the point cloud covers an area 200 m wider than the site in the N, E, S and W direction?
*Buffer is a technical word in geographic information systems that refers to the area that is within a specified distance of a point or surface of interest. The sentence now specifies: "The point cloud extracted on a 200-m radius disk approximately centered on the study site was exported in a single compressed LAS file (v1.1 format 1) [...]"*

- 133: "LAS file (v1.1 format 1)": why different versions (see comment on 131)? Is the LAS version issue of any interest here? If yes, it could be explained here. If not, maybe this would be better formulated in the caption of table 2 or eventually even in a small appendix?

*The format version of files from the data provider (v1.2) was removed from the text. An older format was chosen for the dataset file to ensure compatibility with most software. We think it is useful to specify the version of this file to make it easier for users to identify potential issues when importing the data in their software.*

- 134: "The area located more than 30 m to the SE of the study area was not covered by the acquisition": what should be expressed here, which area? Is there something missing? How does this correspond with the 200 m buffer? This should be made clear here.
*The acquisition did not extend further than 30 m to the southeast of the study area. A part with missing data thus appears in the provided point cloud, but the available area covers the open meadow and the forested site with enough range to compute relevant forest metrics. We think that this information might be confusing for the reader and the sentence was removed.*

- 166: "13 … pyranometers and 11 … pyrgeometers": how does this match what is visible in Figure 1 (11 locations for SW and LW plus 4 locations for SW), shouldn't it be 15 pyranometers? Comparison with the sensor_plan.pdf file (online with the data) suggests that the SW2 and SW3 devices in Figure 1 should be deleted?
*The confusion arises from slight changes in instrumentation between the two field campaigns. The text now mentions 15 pyranometers, as shown in Figure 1 (which is correct).*

- 196: maybe better "until" instead of "through"?
*Changed*

- Figure 6: maybe better make two panels in this figure, one for (i) temperature and snow depth, and a second one for (ii) soil temperatures? Adding a legend with the different colors for the soil temperatures would probably also be a benefit. What about the data gaps in the snow depth time series, where is this explained at least briefly?
*After a few attempts, we feel it's clearer and more concise to show these variables on a single panel. The caption explains the color coding of soil temperature measurements.*
*We added in the caption: "The data gaps in the snow depth time series are mainly due to snowfall events that disrupted the measurements".*

- 208, 211, 213: is the differentiation between snow height and snow depth based on a purpose and carried out consistently all through the text?
*We now use "snow depth" consistently in the text*

- 224: how was the mass of the precipitation tanks measured?
*We have modified this sentence to explain the method more clearly*

- 234: "azimuth angles varying from -90° to +90°": are you sure, where is zero (usually N or S)? Maybe you can add the viewing field of the RLS into Figure 1 (see respective comment there)?
*We added the information that the RLS azimuth angle 0° points towards 225° (south-west)*

- 254: find a better formulation than "uncertainties on the sensors" ("of"?)
*Done*

- 256: "reliable information on sensor accuracy": can you formulate explicitly what the manufacturers generally provide, and replace/delete "information on" (probably it is the accuracy itself which is provided)?
*Table 2 (last column) explicitly states what the manufacturers provide for accuracy estimates.*

- 263: do you mean "relation to the canopy…" instead of "relation with …"?
*Changed*

- 264: "In clear sky": maybe better "In clear sky conditions"
*Changed*

- Figure 10: add a), b), c) and d) in the four panels
*Done*

- 281: what is "0.2°CK"? (probably it is either degree Celsius, or Kelvin)
*Changed to 0.2°C*

- Figure 11: colors in the left panel are orange, red and purple (on the screen), not red and blue as indicated in the text of the caption. Red seems to be the color resulting from the overlay of orange and purple. Printer output may again looks different than screen display. Can one improve this for the sake of clarity?
*For the sake of clarity, we changed the colors in the Figure*

- 300: here would be an appropriate place to add 2-3 sentences about existing modelling studies (and some references) which show how the forest canopy processes mainly depend on LAI, exposition and amount of snow precipitation, amongst other effects. Readers might be interested in this
*Relevant references are already cited in the introduction. More about modelling studies would be out of place in this data paper.*

- Figure 12: add the explicit years to the panels a) and b), and also a complete legend; is it possible to mark the poles such that they can be identified in Figure 1?
*Done*

- Figure 13: add the explicit years to the panels a) and b), and also a complete legend
*Done*

- 311: can table 2 be moved to the end of the text (after "Data availability")?
*Done*

---

## Author Comment (AC2)

Snow accumulation and ablation measurements in a mid-latitude mountain coniferous forest (Col de Porte, France, 1325 m alt.): The Snow Under Forest field campaigns dataset

General comments:

The authors present a much-needed collection of datasets from extensive field measurements to help quantify snow in the canopy at the Col de Porte site. This work will progress our understanding of physical processes in the canopy and provide ground-truth data for future modeling studies that look to improve snow-canopy interactions. In general I though the paper was well written and had very clear methods. I do suggest however that Section 2 seems to encompass a lot of information, and perhaps looking to organize this a bit differently may improve the structure and readability. Also, the instrument uncertainties and the methodology are well documented, but I suggest that more could be stated on post-processing of the data. This includes adding more details on the thresholding approach used in classifying hemispherical photography, more details on leveling/covering correction on radiometer array data, and more details on how snow mass was estimated from the precipitation tanks. I include more details on these points, as well as other points that I believe can help strengthen the paper.

*Thank you for your review. We have taken into consideration all your comments, and a point-by-point response follows.*

Specific comments:

L23 : "This paper presents the field site, instrumentation, and collection methods"… And post-processing?
*Added*

L24: Do you mean to say "an array of radiometers"?
*Changed*

L30: Consider re-working this sentence. You mention the scale of this issue which I feel is important, but I am a bit lost with how you are presenting Rutter et al., 2009 in this context, perhaps finding another reference here would be helpful. Additionally, correct me if I am wrong but this site is not technically classified as boreal forest, which could be misleading to some readers. Finally, is it canopy as a key control on snow cover? The way you have it written seems like you are getting at that snow cover under the canopy influences tree characteristics, and while this has implications for an ecohydrological study, I don't know if it quite captures what you are going for and setting up in this paper.
*The term "boreal" was misleading and has been dropped in this sentence. We have modified this sentence to remove the ambiguities raised by the reviewer.*

L33: Consider removing "e.g.," in your citations it is not needed. This is the same comment throughout
*As numerous studies have been published on this subject and we only cite representative samples, we think that "e.g." is needed here.*

L33: Consider adding "For example," before "The model inter-comparison project SNOWMIP2…"
*Done*

L41: Consider removing "all of the effects of particularly"
*The sentence has been simplified*

L42: In this sentence you should include that detailed canopy measurements are required too?
*This is detailed later.*

L45-47: Consider changing to, "CDP meadow site has been operated by CEN-MeteoFrance, with daily measurements of snow depth, air temperature, and precipitation recorded since 1960 (Lejeune et al., 2019; Morin et al., 2012)."
*Done, thank you for the suggestion*

L61: Same comment as above, consider rewriting to "array of radiatiomers"
*Done*

L65: I feel as if "Site and forest description" does not quite capture all of the work you are doing in this section. Consider revising your sub headings throughout to include the various remote sensing products and snow datasets.
*Indeed, the section "2. Site and forest description" included other work. This has been addressed by creating a separate section "3 Meteorological and snow observations in the forest".*

L69: Consider rewriting to, "The stand is dominated by Norway spruce (Picea abies), with young silver firs (Abies alba) and various broadleaved trees along the western edge of the parcel." …. Also, if you know the types of broadleaved trees it may be useful to include (e.g., x, y, z)..
*This sentence has been rewritten. A table has been added in section 2.2.1 to provide detailed information about tree species and health status.*

L72: While I get how you are displaying the data, this may confuse some readers. Perhaps just write it out. While it is a bit verbose, it will help the reader I'm sure.
*Gaps positions have been added on Figure 2.*

Figure 1: It would be nice to see a location map in the upper right hand corner of Figure 1 to know the approximate location in France. Also, when was the image taken? Perhaps could show time of image on the caption.
*Following the suggestions from reviewer 1, Figure 1 and 2 have been modified and now show the same extent. The figure 1 shows a lot of information and we feel that there is not enough space to add the location in France. The reader can easily locate the study site with the coordinates provided in Introduction. These coordinates are exactly centered on the study site. The time of image has been added in the caption.*

L87: I realize this may seem self-explanatory as why there were particular thresholds for tree heights and DBH, but perhaps one sentence on stating why you made this decision would be

helpful for those that wish to use your canopy data in the future…. Circling back to this after reading ahead, you do this quite a bit, and so adding one sentence on these thresholds would I believe help clear up the next few paragraphs…. i.e., Are you considering trees over 1.3 m because this is the typical snow depth at CDP? Etc…

*Forest inventory practices have different measurement protocols depending on tree size. For large trees, the common variable of interest is the diameter at breast height (usually 1.3 m above ground). The caption of Figure 3 now mentions that trees less than 1.3 m height are not considered in the inventory. We feel that this threshold is relevant as most snow is intercepted by large trees.*

L91: Consider rewriting to "In total, 141 trees were inventoried, including 128 live trees, 3 dead trees and 10 stumps."
*The sentence has been changed.*

L92: If there is a reference for this method, please include it here.
*A reference to Rohle, 1986 has been added. It is also now specified that a clinometer was used to locate the crown extensions.*

L104: It doesn't make sense for the reader here to reference the airborne lidar data that has not been presented yet. I recommend changing Forest inventory to Section 2.2.2 and putting the airborne lidar section first. Then this will allow the reader to understand better in how you are arriving at canopy heights from forest inventory data. This would also allow Figure 2 to make more sense.
*We prefer to begin the forest description with the forest inventory. We think that figure 2 is useful in section 2.1 as a map of the inventoried trees. The technical description of the laser scanning measurements just follows this paragraph.*

L107: Consider changing "southern fence are" to "southern fence included"
*Done*

L113: Is this accuracy the resultant vector of distance and height, or are you just referring to ground surface distance error?
*Assuming an angle error of 1 gradian on the clinometer when checking the vertical position of the observer below the branch extent, the horizontal projection error would be approximately 6 cm for a 4 m tree and 47 cm for a 30 m tree. A few additional centimeters of error can be expected as it is difficult to have the tape measure start exactly at the stem center and end below the observer's eye, while having it perfectly horizontal and tought.*

L122: Provide reference for use of ALS to quantify forest structure, which ever closest matches your method.
*The objective of this data paper is to present the data and how it was acquired and pre-processed. It is not meant to provide a state of the art on the extraction of forest structure metrics relevant for snow interception. The reader can refer to Helbig et al., 2020 for a study on this topic, with details on how the ALS data were processed.*

L134-135: This is unclear as to 1) where exactly this is, and 2) why you are mentioning here
*This sentence has been removed.*

Figure 4: This does not look like the entire 123.5 km2 acquisition, therefore, should specify as "a small subset" or "area of interest". Also, a scale bar and legend for the colorization is needed.
*The figure has been improved and the caption has been changed.*

L153: Maybe to be more concise because I will assume there was no snow on the ground either, you could just say that all sites were snow-free during the hemispherical photographs.
*Done*

L155: An additional sentence on your brightness threshold methodology would be helpful to see here, as well as a reference if you are using an existing method. For example, ss Nobis et al. (2005) mentions there are issues especially in clear sky sun conditions in dense canopy. It would be helpful to see how your method solved for this or adapted to changing cloud cover.
*The brightness threshold methodology is now explained in more detail. The images were not taken in clear sky sun conditions. Nobis et al (2005) has been added as a reference for automatic thresholding, but clear distinctions between canopy and sky were obtained by manual thresholding in every case*

Figure 5: Seems simple but should clearly state in the caption that white pixels are sky and black pixels are canopy if no legend is provided. Alternatively, creating a simple legend could be helpful.
*This information has been added in the caption.*

L169: To what height above the snow were the radiometers typically adjusted to?
*The typical heights are mentioned now*

L175: Please consider expanding on how periods were identified where the sensor was potentially tilted.
*The text mentions that field visits were frequent (every two or three days).*

L178: Quantify the dense canopy. If you have hemispherical photography near this site, I would use these to estimate canopy cover percentage. You could also discuss the CHM measurements at this site. "Rather dense" is a little too arbitrary.
*Unfortunately, we didn't take a hemispherical photograph at the weather station. From photographs taken nearby, we estimate the sky view factor to be around 0.2 at this location (this has been added in the text).*

L183: Check tense in the sentence? Should read "was used"
*Changed to "were used"*

Table 1: Capitalize "variable". Also, consider adding the 15 minute temporal resolution in the table caption. Also, lambda as wavelength is not defined.
*Done*

L196-197: It is a little confusing to read that the temperature gradient of the soil was reversed. Are you saying the temperature gradient within the soil profile matrix reversed, where deeper

soil was warmer? It is a little difficult to see that from this figure 6, but perhaps if you added a Figure 6b that enlarged the snowmelt region you could draw more attention to this.
*We have modified this sentence according to the comment. However, we think that the changes in soil temperature gradients are visible in Figure 6.*

Figure 6: Temperature axis should be black, as there are other temperatures being shown. Having blue axis was confusing at first for me to visualize the results. Also, there are several colors and a legend is needed. Additionally, your Dates September 2018 to June 2019 do not match up with what you present on Line 193.
*Temperature axis is now black and we have changed the dates in the text*

L206: Please change to "were deployed"
*"A transect… was deployed"*

L208: Please add "For each pole the snow height…"
*Done*

L217: Please change to "the forest canopy" or just "the canopy"
*Changed to "the canopy"*

L222: Define CEN
*Changed*

L223: Can shorten here to just "but additional studies are needed."
*Done*

L224: I'm not sure if I am catching them all but watch out for this where you are using "was" instead of "were". Sentence should be changed to "precipitation tanks were measured…"
*"The mass... was measured"*
*The sentence was modified following a comment from referee 1.*

L224: Additionally, it is not clear how the "mass of snow" was collected? Are you making assumptions for density? Are the precipitation tanks measuring weight? Are there markers on the tanks to know the depth?
*As the mass was measured, there is no need to make assumptions about snow density or to document snow depth in the tanks. We specify in the revised text that the snow was taken from the tanks and weighed.*

L236: You can shorten this down and just say that, "Data acquired during a given day by the laser meter is interpolated averaged on a regular 3-cm grid".
*We think that all the information provided by this sentence is required.*

L242: change "has" for "had"
*Done*

Figure 8: Please be consistent with either snow height (as shown in the legend) or snow depth as shown in the caption throughout the text for clarity. Either are fine, but it would be clearer

to stay with just one. Also, make sure to use past tense on your verbs throughout, "The measurement area encompassed three radiometers (I3, I4, and I5)."
*The caption has been modified*

L264: Sunflecks as one word
*Done*

L265: Please remove the rest of the sentence after ",superimposed". This does not need to be stated I believe.
*We think it is important to note the contrast between the strong spatial variations of direct shortwave radiation below the canopy (sunflecks) and the more spatially constant diffuse shortwave radiation that penetrated through the canopy.*

L267: Are you referring to Figure 10 C or D here?
*10d, now specified*

L268: Once again I feel like you are stating something that doesn't need to be there. The added longwave radiation is from the forest and is well known. Additionally, please check grammar on this sentence. Colons should be followed by a list, not a statement.
*Colons have been changed to semicolons*
*We think it is useful to emphasize the remarkably constant offset between long-wave radiation signals. However, if the referee so requests, we can delete this sentence.*

Figure 10: Line color for sky view factor needs to be shown in a legend. Additonally, you need to have a, b,c, and d marked on the figure.
*a, b,c, and d have been added in the figure. We feel that the legend clearly describes the line colors and that it is therefore not necessary to add a legend (in agreement with reviewer 1).*

L280: Once again, check you are using the correct use of colons. Please consider changing to "Thus, average daily air temperatures are quite similar in the forest and meadow site (dT ~ 0.2°CK on average during the 2018 campaign) because warmer nights counterbalance cooler days in the forest relative to the meadow. "
*Sentence changed, thank you for the suggestion*

L281: Celsius (C)?
*Changed*

Figure 11: Capitalize "Difference" and also please provide a legend for day time versus night time measurements. Also, please provide (A) and (B) on the figure itself.
*The figure and the caption have been changed following the suggestion.*

L296-299: The ablation is a little more nuanced and for that I agree that more research needs to be done. However, I recommend reading Jessica Lundquist's meta-analysis for looking at the ablation melt phase and how this varies across snow regimes and climate. I believe it would be a good idea to provide context on how your field data relates to this meta-analysis.
*Thank you for this suggestion. A reference to Lundquist et al. (2013) has been added. Certainly, more research is required to investigate this point at Col de Porte.*

Table 2: Should this be listed instead in the code and data availability statement?
*Following a suggestion from referee 1, Table 2 has been moved after Data availability*